# HENASY: Learning to Assemble Scene-Entities for Interpretable Egocentric Video-Language Model

**Khoa Vo     Thinh Phan     Kashu Yamazaki     Minh Tran     Ngan Le**
AICV Lab, University of Arkansas, Fayetteville, USA
{khoavoho,thinhp,kyamazak,minht,thile}@uark.edu

## Abstract

Current video-language models (VLMs) rely extensively on instance-level alignment between video and language modalities, which presents two major limitations: (1) visual reasoning disobeys the natural perception that humans do in first-person perspective, leading to a lack of reasoning interpretation; and (2) learning is limited in capturing inherent fine-grained relationships between two modalities.

In this paper, we take an inspiration from human perception and explore a compositional approach for egocentric video representation. We introduce *HENASY (Hierarchical ENtities ASsemblY)*, which includes a spatiotemporal token grouping mechanism to explicitly assemble dynamically evolving scene entities through time and model their relationship for video representation. By leveraging compositional structure understanding, HENASY possesses strong interpretability via visual grounding with free-form text queries. We further explore a suite of multi-grained contrastive losses to facilitate entity-centric understandings. This comprises three alignment types: video-narration, noun-entity, verb-entities alignments.

Our method demonstrates strong interpretability in both quantitative and qualitative experiments; while maintaining competitive performances on five downstream tasks via zero-shot transfer or as video/text representation, including video/text retrieval, action recognition, multi-choice query, natural language query, and moments query.

Project page: https://uark-aicv.github.io/HENASY

## 1 Introduction

Recent advancements in technology and hardware devices for augmented reality (AR) have fueled hopes for virtual assistant applications that can provide users a wide range of assistance, such as real-time procedural instructions, moments retrieval, and interactive learning experiences, all through egocentric video streams of similar perspective with user. Publicly available massive-scale egocentric datasets such as Ego4D [1] and Epic Kitchens-100 [2], providing suites of egocentric tasks, have further sparked even more interest within the research community.

Video-language models (VLMs) have currently become a *de-facto* approach to egocentric video understanding. By learning robust visual-language representations from video-caption pairs [3], VLMs can be applied flexibly to a wide range of downstream tasks, either through zero-shot transfer or as modality encoders. Existing state-of-the-art (SOTA) VLMs for egocentric videos [4, 5, 6, 3] exhibit remarkable performances by following CLIP-like [7] dual-encoder architecture. During training, these models generally learn through the *instance-level* alignment [8, 3] between pairs of video and caption representations (Fig. 1(a)).

However, videos consist of complex and dynamic interactions among arbitrary entities, which cannot be effectively captured by simple *instance-level* alignment alone. In fact, a caption contains textual elements that concisely capture video entities. For examples, nouns indicate entity occurrences [4],

38th Conference on Neural Information Processing Systems (NeurIPS 2024).

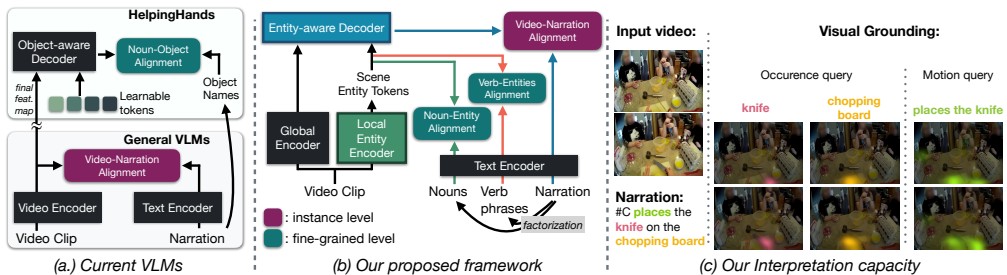

Figure 1: **Problem Overview. (a)** Current VLMs [5] rely on instance-level contrastive learning between video & narration. HelpingHands [4] implicitly induces object occurrence information into video features at final layer of video encoder. **(b)** Our proposed (*HENASY*) aims to assemble dynamic entities from video patches via *local entity encoder*, while *entity-aware decoder* captures interactions between entities and global context to form comprehensive video. HENASY is trained with suite of multi-grained contrastive alignments to enforce visual representations entity-level upto video-level. **(c)** By such compositional approach, HENASY is the first VLM that shows strong interpretability via visual grounding with both appearance/motion query types.

while verb phrases convey motion information [9] in the video. To fully capture these fine-grained alignments, a VLM will perform more effectively if it: (1) understand videos in a bottom-up manner, where semantically similar patches form entities, and relationships between entities construct the video representation; and (2) explicitly model fine-grained relationships between video entities and nouns/verbs to comprehensively capture appearance/motion information, respectively.

Human perception aligns closely with the above requirements. We perceive the dynamic surroundings in a compositional manner [10], where distinct entities emerge from smaller parts that combine to form a whole. Each entity maintains spatial and temporal coherence and interacts with others only when in close proximity. Understanding the compositional structure of the surroundings enables us to intrinsically comprehend and memorize information, while also allowing us to provide *interpretations* of our decision-making process, which is absent in current egocentric VLMs.

Inspired by such observation, we propose *HENASY: Hierarchical ENtities ASsemblY* framework (pronounced heh-nuh-see), which follows compositional understanding as in Fig 1(b). Concretely, HENASY comprises three key components: *(i) Local Entity Encoder*, a hierarchical transformer-based encoder that learns to assemble dynamic scene-entities from video patches via our proposed spatiotemporal token grouping mechanism, which is an enhanced version from slot-based groupings in stationary images [11, 12]; *(ii) Global Encoder*, a pre-trained video representation module that perceives the input video at a global level; and *(iii) Entity-Aware Decoder*, which models the internal interactions among scene entities and their relationship with the global features, thereby enriching the entity-centric video representation extraction. Furthermore, HENASY is able to perform visual grounding to obtain dynamic segmentations corresponding to either entity or activity with the produced entity embeddings and their attention maps as a side product of its local entity encoder, showing promising interpretation via dynamic saliency maps across frames (Fig. 1(c)).

Developing an effective model necessitates a strong network architecture and well-defined objectives. With the proposed HENASY architecture, instance-level contrastive loss only handles global alignment, failing to address dynamic entity alignment. Hence, we introduce multi-grained contrastive losses to optimize HENASY for both entity- and video-level representations using narration alone. Specifically, HENASY is trained with three types of alignment: video-narration, noun-entity, and verb-entities. While the first two employ instance-level contrastive loss and model object occurrences via narration's nouns, respectively, verb-entity alignment is newly introduced. It aims to incorporate activity/motion information from narration's verb phrases into entities using a *one-to-many strategy*, which emphasizes the alignment of a verb phrase to the most semantically relevant entities. Additionally, we propose a new *projection loss* that employs detected hand/object boxes [4] to ensure segmentation masks tightly cover respective entities, enhancing HENASY's interpretative robustness.

We are the first to demonstrate the value of compositional perception approach for egocentric video understanding. Our experiments show that by tasking our proposed *local entity encoder* to assemble dynamic entities, video representations are effectively improved to outperform current VLMs in a wide range of benchmarks, including video retrieval (EgoMCQ [3] & EpicKitchen-MIR [2]),

activity recognition (EpicKitchen-CLS & EGTEA [13]) via zero-shot transfer. Furthermore, temporal localization models [14, 15] equipped with HENASY video/text features can achieve state-of-the-art performances in episodic memory tasks of EgoNLQ and EgoMQ [1]. Finally, HENASY possesses strong interpretability that is quantitatively and qualitatively superior to current VLMs.

## 2 Related Works

**Video-Language Pre-Trained Models.** Pre-training VLMs on a large-scale dataset of video-text pairs and deploying them in down-stream tasks has now become a standard practice. Transformer-powered pre-trained VLMs [16, 17, 18, 19, 3, 6, 5, 4, 9] have accomplished superior results on a wide range of tasks, i.e., text-to-video retrieval, action recognition, or events localization. VLMs can be divided into two common categories, i.e., unified- and dual-encoder. The former models [16, 18, 20] fuse multimodal via cross-attention and can be trained with proxy tasks of masked language modeling [21, 22] or masked frame modeling [18, 16]. The latter models [23, 3, 5, 6, 9] employ separate encoders for video and text, trained jointly via contrastive learning [8, 3].

Recently, several VLMs [4, 17, 9] employ fine-grained information from captions by decomposing them to capture object/activity through nouns/verbs, respectively. However, these models do not fully exploit fine-grained learning within the video encoder itself, and true granular-level alignment between modalities remains unexplored. *In our work, we explicitly model visual content as dynamic entities, capturing their interactions to form a comprehensive video representation. Additionally, our proposed method is trained with multi-grained objectives, ranging from video-text, noun-entity, to verb-entities pairs.*

**Interpretable Video Representation.** There have been efforts to enhance the interpretability of video representations [24, 25, 26, 27], these typically involved factorizing videos into entities and environmental contexts, and utilizing adaptive attention mechanisms [24] to selectively focus on relevant entities. Such mechanisms enable interpretation through their selection module, highlighting only the primary entities contributing to the final predictions. However, these approaches often require entities to be pre-detected by external models, which is restricted to a predefined set of objects. Additionally, these works have focused on models specifically designed for individual tasks, e.g., temporal action detection, video dense captioning. *In contrast, our work introduces an end-to-end method capable of learning to form entities without an off-the-shelf detector. Moreover, we aim to develop a robust video-language model (VLM) that is versatile across a variety of tasks.*

**Interpretation via Object Discovery.** Recent years have seen a growing body of research in end-to-end learning of object discovery, which learns to decompose an image or a video into distinct objects without direct supervision. Slot-based methods such as IODINE [28] and Slot Attention [11] utilize mixture-model likelihoods [29] to tackle this challenge, demonstrating promising performance through evaluations on synthetic images with simple objects. Subsequently, GroupViT [12] and ODIN [30] incorporate slot attention with contrastive learning and achieved notable success in identifying semantic groupings on natural in-the-wild images. However, these models are not capable of modeling dynamic objects in videos domain. To mitigate this problem, SaVI++ [31] proposes a workaround technique, which requires groundtruth depth information in a reconstruction objective to bootstrap object discovery training. In our work, we enhance slot-based grouping mechanisms introduced in GroupViT [12] to model temporal coherency of dynamic objects in videos. *Different from [31], HENASY does not require any extra data further than color video sequences. Instead, HENASY utilizes learned patch features of pre-trained global encoder to bootstrap several early layers of its local entity encoder for entities grouping via a cross-attention mechanism.*

## 3 Preliminaries

**Video-language representation learning** aims to learn a common latent space to represent video and text. A training dataset for this task comprises of $N$ tuples $\{\mathcal{V}_i, \mathcal{T}_i\}_{i=1}^N$, where $\mathcal{V}_i$ denotes a short sequence of RGB frames, and $\mathcal{T}_i$ is a free-form text sentence that describes visual content.

**Dual-encoder architecture** is a common paradigm that current SOTA VLMs [4, 5, 3] employ for the above task, which consists of (a) a visual encoder $f$ mapping the input video $\mathcal{V}_i$ to a visual embedding feature $\mathbf{v}_i = f(\mathcal{V}_i)$, and (b) a language encoder $g$ mapping the text $\mathcal{T}_i$ to a linguistic embedding feature $\mathbf{t}_i = g(\mathcal{T}_i)$.

**Contrastive-based losses** are common objective for video-language representation. Given a batch of $B$ normalized video-text embedding pairs $i = \{\hat{\mathbf{v}}_i = \mathbf{v}_i/|\mathbf{v}_i|, \hat{\mathbf{t}}_i = \mathbf{t}_i/|\mathbf{t}_i|\}$, a contrastive-based loss pulls embeddings of aligned (positive) pairs close in feature space, while pushing embeddings of misaligned (negative) pairs away. We adopt *EgoNCE* variation [3] of contrastive loss as one of our training objectives because of its effective approach in identifying positive and negative pairs. Specifically, each sample $i \in B$ is associated to a set of positives $P_i$ constructed by comparing nouns and verbs across all texts. Additionally, for each sample $i$, a hard negative $i'$ is sampled from a temporally adjacent segment within the same video, expanding our batch to $\widetilde{B}$. For a more in-depth discussion on the strategy used for sample selection, refer to [3]. Herein, the video-to-text objective is succinctly expressed as:

$$\mathcal{L}_{ego}^{v2t} = \frac{1}{\widetilde{B}} \sum_{i \in \widetilde{B}} \log \frac{\sum_{p \in P_i} \exp(\hat{\mathbf{v}}^T \hat{\mathbf{t}}_p / \tau)}{\sum_{n \in B} \exp(\hat{\mathbf{v}}^T \hat{\mathbf{t}}_n / \tau) + \exp(\hat{\mathbf{v}}^T \hat{\mathbf{t}}_{n'} / \tau)} \quad \text{where } \tau \text{ denotes a temperature} \quad (1)$$

The objective of text-to-video $\mathcal{L}_{ego}^{t2v}$ is derived from Eq. 1 by inverting $v$ and $t$, and EgoNCE loss is a summation of both directions $\mathcal{L}_{ego} = \mathcal{L}_{ego}^{v2t} + \mathcal{L}_{ego}^{t2v}$.

## 4 HENASY

We present the HENASY framework (Fig. 2) for egocentric video-language modeling. HENASY is a compositional video understanding approach featuring a dual-encoder architecture, designed to explore an interpretable, entity-based visual representation. Specifically, besides typically capturing global features via global encoder (Sec. 4.1), our video encoder also assembles dynamic scene entities from video patches via *local entity encoder* (Sec. 4.2), then *entity-aware decoder* (Sec. 4.3) models their intra-connections as well as inter-connections with global features to form a comprehensive video representation. Our objective is to develop an interpretable reasoning process that robustly supports decision-making, while allowing visual grounding with text queries. To achieve this target, it requires not only an effective network design, but also a suite of multi-grained contrastive learning (Sec. 4.4) to enforce both entity-level and video-level representations.

For readability, we denote five types of tokens as follows: $\mathbf{z}$ for video patch tokens, $\mathbf{c}$ for learnable video tokens, $\mathbf{g}$ for learnable group tokens, $\mathbf{s}$ for segment tokens, and $\mathbf{e}$ for entity tokens.

### 4.1 Global Encoder

Global encoder provides global visual information of the entire input video. We adopt the pre-trained TimeSFormer [32] to capture the global visual context of the entire input video. Particularly, we follow the protocol of [3, 5, 4] to decompose the given input video sequence $\mathcal{V}_i \in \mathbb{R}^{T \times 3 \times H \times W}$ with $T$ RGB frames or resolution $H \times W$ into $T \times K$ non-overlapping patches of resolution $P \times P$, where $K = HW/P^2$. Then, every patch is linearly projected by a 2D convolutional layer, forming video patch tokens $\mathbf{z} \in \mathbb{R}^{TK \times D}$ (with $TK = T \times K$) representing embedding features at every temporal and spatial location, where $D$ is hidden dimension. TimeS-Former processes the video patch tokens $\mathbf{z}$ with an additional learnable video tokens $\mathbf{c} \in \mathbb{R}^{1 \times D}$ through a stack of divided space-time attention (*DST*) blocks, which is described as follows: $[\mathbf{c}^{l+1}; \mathbf{z}^{l+1}] = \text{DST}([\mathbf{c}^l; \mathbf{z}^l])$ where $[\cdot; \cdot]$ denotes concatenation operator. Please see Appendix A for more details on the computational process of a *DST* block.

### 4.2 Local Entity Encoder

Local entity encoder models fine-grained information of the input video by consistently capturing dynamic scene-entities. We adopt a *hierarchical bottom-up architecture*, consisting of attention-based layers that are divided into multiple stages. Each stage progressively groups small video patch tokens $\mathbf{z}$ from the previous stage into larger segments. As a result, local entity encoder forms scene entity tokens that depict individual entities, which dynamically evolve across video frames.

While following GroupViT [12] to directly process input video patch tokens could be an option, we find doing that diminishes performance. Additionally, the tokens grouping mechanism from [12, 11] is not capable of modeling dynamic entities in videos domain. To address these challenges, we introduce a *bootstrapping stage*, which couples itself with early layers of the global encoder through

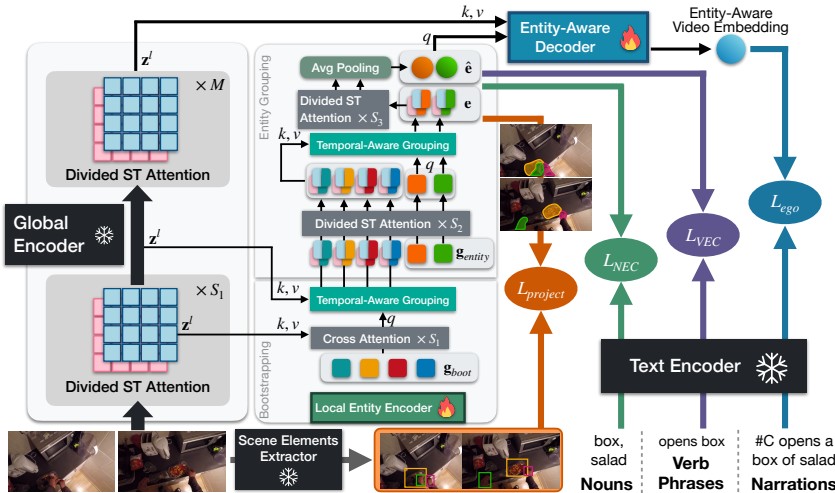

Figure 2: **Overview of the HENASY framework for video-language modeling. Left:** HENASY features a dual-encoder architecture with a compositional video understanding approach. The local entity encoder assembles dynamic scene entities from video patches, while the global encoder provides contextual features. These are combined in the entity-aware decoder to create an interpretable video representation. **Right:** HENASY is supported by a suite of multi-grained contrastive learning to enforce both entity-level and video-level representations.

cross-attention to group video patches into initial entities' segments. Furthermore, to capture dynamic entities in video, we introduce *temporal-aware grouping* mechanism.

**Bootstrapping Stage.** Bootstrapping stage consists of $S_1$ consecutive cross-attention layers, which takes a set of learnable group tokens $\mathbf{g}_{boot}^l \in \mathbb{R}^{G \times D}$ as initial queries ($G$ is the initial number of tokens). At each cross-attention layer $l$ starting from 0, the queries aggregate information from the patch tokens $\mathbf{z}^l$ at corresponding layer of global encoder:

$$\mathbf{g}_{boot}^{l+1} = \texttt{CrossAtt}^l(\mathbf{g}_{boot}^l, \mathbf{z}^l), \quad \text{for } l = 0, .., S_1 - 1 \tag{2}$$

At the final layer of bootstrapping stage, we obtain $\mathbf{g}_{boot}^l$. This is then associated with patch tokens $\mathbf{z}^l$ from the corresponding layer of global encoder within *temporal-aware grouping* block (TAG) to group patches into larger segment tokens:

$$\mathbf{s}^l = \texttt{TAG}(\mathbf{g}_{boot}^l, \mathbf{z}^l) \quad , \text{where } l = S_1 \tag{3}$$

Herein, $\texttt{TAG}(\mathbf{q}, \mathbf{k})$ merges key tokens $\mathbf{k}$ together based on their similarities with query $\mathbf{q}$, while preserving the temporal dimension of key tokens (discussed in detail later). As a result, we obtain new segment tokens $\mathbf{s}^l \in \mathbb{R}^{TG \times D}$, which is then utilized as inputs to the *entity grouping stage*.

**Entity Grouping Stage.** From this point, local entity encoder is decoupled from the global encoder and is trained to merge these input segments $\mathbf{s}$ into complete scene entities $\mathbf{e}$. At this stage, a new set of learnable group tokens $\mathbf{g}_{entity}^l \in \mathbb{R}^{E \times D}$ is introduced, which aims to relate segment tokens with similar semantics into an individual scene entity. It is important to note that maintaining consistent temporal dynamics is required at each stage. Therefore, we adopt $S_2$ DST blocks [32] to propagate information mutually between learnable group tokens and segment tokens: $[\mathbf{g}_{entity}^{l+1}; \mathbf{s}^{l+1}] = \texttt{DST}([\mathbf{g}_{entity}^l; \mathbf{s}^l])$, for $l = S_1, .., S_1 + S_2 - 1$.

After the final layer, segment tokens $\mathbf{s}^l$ are grouped to generate intermediate entity tokens, i.e., $\hat{\mathbf{e}}^l = \texttt{TAG}(\mathbf{g}_{entity}^l, \mathbf{s}^l) \in \mathbb{R}^{TE \times D}$, where $l = S_1 + S_2$. Then, to enable interactions between scene entities and across temporal dimension, we apply a stack of $S_3$ DST blocks to all entity tokens: $\hat{\mathbf{e}}^{l+1} = \texttt{DST}(\hat{\mathbf{e}}^l)$, for $l = S_1 + S_2, .., S_1 + S_2 + S_3 - 1$.

We observed that segment tokens $\mathbf{s}$ and entity tokens $\mathbf{e}$ facilitate temporal consistencies within the temporal attention of a DST block. Unlike in TSF [32], where tokens are spatially limited within a patch, these tokens can evolve freely across frames, enhancing the flexibility of the time-attention mechanism in a DST block. To obtain spatio-temporal entity embeddings, we apply temporally

average pooling on entity tokens of the final layer: $\mathbf{e} = \texttt{AvgPool}(\hat{\mathbf{e}}^l)$, where $\mathbf{e} \in \mathbb{R}^{E \times D}$ and $l = S_1 + S_2 + S_3$.

**Temporal-Aware Grouping (TAG).** As aforementioned, TAG is employed at the final layers of bootstrapping stage and entity grouping stage, to merge semantically similar tokens (i.e., $\mathbf{z}$ or $\mathbf{s}$) into a larger segment while preserving the temporal dimension. Generally, this mechanism takes a set tokens $\mathbf{i} \in \mathbb{R}^{TI \times D}$ ($\mathbf{i}$ can be either $\mathbf{z}$ or $\mathbf{s}$) as inputs and a set of learnable group tokens $\mathbf{g}_q \in \mathbb{R}^{Q \times D}$ as queries. We re-shape $\mathbf{i}$ into 3-dimension $\mathbb{R}^{T \times I \times D}$ and evaluate the similarity between $\mathbf{g}_q$ and $\mathbf{i}$.

It firstly evaluates similarity between each group token and every input token, forming a 3D similarity matrix $\mathbf{A} \in \mathbb{R}^{T \times Q \times I}$. Then, an assignment matrix $\tilde{\mathbf{A}} \in \{0,1\}^{T \times Q \times I}$ is computed, assigning each input token to the most relevant group based on similarity scores. Finally, it performs per-frame groupings, merging the input tokens of the same group together, forming a set of new tokens $\mathbf{o} \in \mathbb{R}^{T \times Q \times D}$ representing larger segments. We formalize the computation of every new group as follows:

$$\mathbf{o} = \texttt{TAG}(\mathbf{g}_q, \mathbf{i}) ; \quad \mathbf{o}_{t,i} = \texttt{TAG}_{t,i}(\mathbf{g}_q, \mathbf{i}) = (\mathbf{g}_q)_i + \frac{\sum_{j=1}^{I} \tilde{\mathbf{A}}_{t,i,j} \mathbf{i}_{t,j}}{\sum_{j=1}^{I} \tilde{\mathbf{A}}_{t,i,j}} \tag{4}$$

Afterwards, we re-shape $\mathbf{o}$ to $\mathbb{R}^{TQ \times D}$ as output of TAG.

The computation of similarity matrix $\mathbf{A}$ and assignment matrix $\tilde{\mathbf{A}}$ are detailed in Appendix B.1. Additionally, the derivation of saliency maps for interpreting assignments between video patches and entities is explained Appendix B.2.

### 4.3 Entity-Aware Decoder

We seek to propagate entity-level features $\mathbf{e}^l$ from the local entity encoder to the final video embedding for a comprehensive video representation. For this purpose, we introduce *entity-aware decoder*, which is illustrated in Fig. 3. Entity-aware decoder includes a stack of hybrid-attention blocks to refine the interactions between entities and video patches, and render the video embedding. At a block $b_{dec}$, it first performs cross-attention with entity tokens as queries and patch tokens as keys, values. Then, self-attention followed by a multi-layer perceptron (MLP) is applied over the output:

$$\tilde{\mathbf{e}}^{b_{dec}} = \texttt{CrossAtt}(\mathbf{e}^{b_{dec}}, \mathbf{z}^l)$$
$$\mathbf{e}^{b_{dec}+1} = \texttt{MLP}\big(\texttt{SelfAtt}(\tilde{\mathbf{e}}^{b_{dec}})\big) \tag{5}$$

Eventually, we obtain the video representation, dubbed as entity-aware video embedding $\mathbf{v}$, by averaging the final outputs of entity tokens:

$$\mathbf{v} = \texttt{AvgPool}(\mathbf{e}^{b_{dec}}) \tag{6}$$

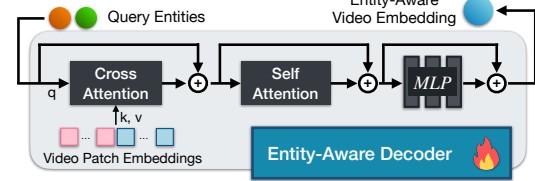

Figure 3: Illustration of entity-aware decoder.

### 4.4 Multi-grained Contrastive Learning

Beside video-narration contrastive loss [8, 3], which captures coarse-grained semantic alignment between the video and narration, we introduce two finer-grained contrastive losses: noun-entity contrastive loss (NEC) and verb-entities contrastive loss (VEC), which focuses on inducing visual appearance and motion cues directly to the composed entities. We also utilize projection loss, leveraging object boxes from an off-the-shelf detector [33] as a weak supervision to encourage the generated entity masks tightly conforming to the corresponding entity, promoting robust interpretability of our proposed model.

**Noun-Entity Contrastive Loss (NEC).** From the groundtruth narration, we obtain $N_n$ nouns and their embeddings $\mathbf{n} \in \mathbb{R}^{N_n \times D}$ via the text encoder. Following [4], we compute a similarity matrix between noun embeddings and entity embeddings. Every noun is matched with an entity token having highest similarity score via Hungarian matching. Following this, we construct a noun-entity contrastive loss using the InfoNCE [8], where positive pairs consist of the matched noun embedding $\mathbf{n}_p$ and entity embedding $\mathbf{e}_p$. The contrast is defined over the embeddings $\mathbf{n}'_j$ of all nouns in the dataset taxonomy dictionary $\mathcal{D}$ [1]:

$$\mathcal{L}_{NEC} = -\frac{1}{N_n} \sum_{p=1}^{N_n} \log \frac{\exp(\mathbf{e}_p^T \mathbf{n}_p / \tau)}{\sum_{j \in \mathcal{D}} \exp(\mathbf{e}_p^T \mathbf{n}'_j / \tau)} \tag{7}$$

**Verb-Entities Contrastive Loss** is a new loss term that instills motion information directly into entity tokens from narration's verb phrases. As suggested in [9] that LLMs are superior to classical methods such as part-of-speech tagging in retrieving verb phrases, we use a Llama-2 [34] to obtain $N_v$ verb phrases from an input narration. Given that a verb phrase describes an activity involving several scene entities, we introduce *weighted many-to-one alignment* strategy to prioritize the most relevant entity-verb alignments. Firstly, let $\mathbf{a}_i \in \mathbb{R}^D$ be one of the embedded verb phrases, we obtain a Softmax-normalized similarity scores between $\mathbf{a}_i$ and every entity $\mathbf{e}_j$: $s(\mathbf{a}_i, \mathbf{e}_j) = \frac{\mathbf{a}_i \cdot \mathbf{e}_j^T}{\sum_k^E \mathbf{a}_i \cdot \mathbf{e}_k^T}$. Then, we re-weight entities by the computed weight and obtain a weighted average of entities representation: $\mathbf{e}^{avg} = \sum_j^E s(\mathbf{a}_i, \mathbf{e}_j)\mathbf{e}_j$. Here, $\mathbf{e}^{avg}$ re-weights each entity based on its relevancy with verb phrase $\mathbf{a}_i$. Finally, we compute contrastive loss between this paired representations:

$$\mathcal{L}_{VEC} = -\frac{1}{N_v} \sum_{p=1}^{N_v} \log \frac{\exp\left((\mathbf{e}_p^{avg})^T \mathbf{a}_p / \tau\right)}{\sum_{j \in B} \exp\left((\mathbf{e}_p^{avg})^T \mathbf{a}_j / \tau\right)} \tag{8}$$

where we utilize batch formation technique from egocentric contrastive loss [3] to form negatives set in $\mathcal{L}_{VEC}$.

**Projection Loss** operates on each individual frame of the input video, utilizing an external object detector [33] to identify bounding boxes $b = \{b_i \in \mathbb{R}^4\}_{i=1}^{N_b}$ of scene entities. Let $\mathbf{m} = \{\mathbf{m}_i \in (0,1)^{H \times W}\}_{i=1}^E$ be the predicted saliency maps of scene entities (see Appendix B.2 for saliency maps formulation), Hungarian matching pairs each detected box $b_i$ with the predicted mask $\mathbf{m}_i$ having the highest IoU.

Designing a differentiable loss function that guides the predicted mask $\mathbf{m}_j$ by groundtruth box $b_j$ is quite challenging. To address this, we utilize an axis projection function [35] to minimize the discrepancy of vertical and horizontal projections of $b_j$ and $\mathbf{m}_j$ on two axes. This ensures that the smallest box encompassing $\mathbf{m}_j$ matches with $b_j$. Concretely, $b_j$ is firstly converted to binary mask format $\hat{\mathbf{b}}_j \in \{0,1\}^{H \times W}$ where pixels inside $b_j$ is assigned by 1 and 0 otherwise. Then, a projection loss is defined as follows:

$$\mathcal{L}_{proj} = \frac{1}{N_b} \sum_{j=1}^{N_b} \left( \mathcal{L}_{dice}\big(\max_y(\mathbf{m}_j), \max_y(\mathbf{b}_j)\big) + \mathcal{L}_{dice}\big(\max_x(\mathbf{m}_j), \max_x(\mathbf{b}_j)\big) \right) \tag{9}$$

where $\mathcal{L}_{dice}$ is a Dice loss function [36], $\max_y(\cdot)$ and $\max_x(\cdot)$ are max-project operators along $y$-axis and $x$-axis of the frame, respectively.

**Total Optimization.** Overall, our model is optimized with a weighted sum of EgoNCE loss over video-text pairs and three objectives stated above:

$$\mathcal{L} = \mathcal{L}_{ego}^{v2t} + \mathcal{L}_{ego}^{t2v} + \lambda_1 \mathcal{L}_{NEC} + \lambda_2 \mathcal{L}_{VEC} + \lambda_3 \mathcal{L}_{proj} \tag{10}$$

where $\lambda_1$, $\lambda_2$, and $\lambda_3$ balance contributions of different loss terms.

## 5 Experiments

### 5.1 Training and Implementation Details

**Architecture.** We use video clip inputs of size $224 \times 224$, text inputs are tokenized and processed by a 12-layer Transformer following [5]. We employ TimeSFormer [32] Base (TSF-B) for the global encoder. In the local entity encoder, all layers share a hidden dimension $D = 512$. Bootstrapping stage includes $S_1 = 6$ cross-attention layers with 64 group tokens. Entity grouping stage consists of $S_2 = 3$ DST blocks with 8 group tokens, followed by $S_3 = 3$ DST blocks. Entity-aware decoder is a stack of 3 hybrid-attention blocks.

**Training.** HENASY is trained on EgoClip [3], which contains 3.8M clip-narration pairs covering a sub-set of 2,927 video hours from Ego4D [1]. For each video clip, we uniformly sample 4 frames. We employ the pre-extracted narration's nouns and pre-detected hand and object bounding boxes from [4] for NEC loss and projection loss, respectively. For verb phrases, we employ Llama-2 [34] with a prompt as discussed in Appendix C. The loss weights in Eq. 10 are set as: $\lambda_1 = 0.5, \lambda_2 = 0.5, \lambda_3 = 1.0$. We train HENASY on two A6000 GPUs, in 5 epochs with AdamW optimizer [37] at fixed learning rate of $3e-5$, and with batch size of 128. We initialize global encoder and text encoder with pretrained model provided from [5], but freeze them in the entire training process.

Table 1: Comparison on the zero-shot transfer over EgoMCQ, EK100-MIR, EK100-CLS, and EGTEA. HelpingHands* refers to our re-produced results with TSF-B backbone using provided source code [4] .

| Methods | EgoMCQ | | EK100-MIR | | | | | | EK100-CLS | | EGTEA | |
|---|---|---|---|---|---|---|---|---|---|---|---|---|
| | | | mAP | | | nDCG | | | Top-1 | Top-5 | Top-1 | Mean |
| | Inter | Intra | V-T | T-V | Avg | V-T | T-V | Avg | Acc | Acc | Acc | Acc |
| EgoVLP [3] | 90.6 | 57.2 | 26.0 | 20.6 | 23.3 | 28.8 | 27.0 | 27.9 | - | - | 17.6 | - |
| EgoVLPv2 [6] | 91.0 | 60.9 | - | - | 26.7 | - | - | 29.1 | - | - | - | - |
| LaViLa [5] | 93.8 | 59.9 | 35.1 | 26.6 | 30.9 | 33.7 | 30.4 | 32.0 | 16.4 | 34.4 | 35.5 | 28.9 |
| HelpingHands* [4] | 93.2 | 58.8 | 35.6 | 26.8 | 31.2 | 34.7 | 31.7 | 33.2 | - | - | 35.3 | 29.4 |
| **Ours** | **94.1** | **61.3** | 35.5 | **27.1** | **31.3** | 34.6 | **31.7** | **33.2** | **19.5** | **38.2** | **35.9** | **29.6** |

Table 2: Comparison on the visual & textual representation over EgoNLQ and EgoMQ. Grey indicates result we obtained using provided pre-trained checkpoint that.

| Methods | EgoNLQ | | | | EgoMQ | | |
|---|---|---|---|---|---|---|---|
| | mIoU@0.3 | | mIoU@0.5 | | R1@0.5 | R5@0.5 | mAP |
| | R1 | R5 | R1 | R5 | | | |
| SlowFast [38] | 5.5 | 10.7 | 3.1 | 6.6 | 25.2 | 46.2 | 6.0 |
| EgoVLP [3] | 10.8 | 18.8 | 6.8 | 13.5 | **30.1** | **52.0** | 11.4 |
| LaViLa(B) [5] | 10.5 | 19.1 | 6.7 | 13.6 | 27.4 | 49.0 | 11.3 |
| HelpingHands* [4] | 11.2 | 20.4 | 6.9 | **14.7** | 27.5 | 49.0 | 11.7 |
| **Ours** | **11.5** | **21.5** | **7.0** | **14.7** | 28.3 | 51.0 | **12.4** |

## 5.2 Benchmarks and Evaluation Protocols

**Ego4D benchmarks [1].** Ego4D is the largest publicly available egocentric video dataset, featuring 3,670 hours of daily-life activity video for a wide range of benchmarks. We evaluate on three tasks:

- *EgoMCQ* [3]: A multi-choice questions task to select the correct video clip from 5 candidates for each query. Accuracy is evaluated in intra-/inter-video (candidates from the same/different video).
- *EgoNLQ*: A sub-task in episodic memory involving localizing video intervals that answer a given a free-form text query. Evaluation metrics include Recall@$K$ for mIoU thresholds $\theta$, where $K \in \{1, 5\}$ and $\theta \in \{0.3, 0.5\}$.
- *EgoMQ*: Also a sub-task of episodic memory, it involves identifying and categorizing action instances from 110 activity classes. Evaluation metrics are recalls (at mIoU=0.5) and mean Average Precision (mAP).

**EpicKitchens 100 benchmarks [2]**: This dataset focuses on indoor and kitchen activities with 100 hours of video. We evaluate two tasks:
- *EK100-MIR*: A multi-instance retrieval task evaluating video and narration matching in both T→V and V→T. Metrics are mAP and normalized Discounted Cumulative Gain (nDCG).
- *EK100-CLS*: A action recognition task classifying videos into 300 noun classes or 97 verb classes. Metrics are Top-1 and Top-5 accuracy.

**EGTEA benchmark [13]**: This dataset contains 28 hours of video with 106 classes. We evaluate fine-grained cooking action recognition *EGTEA* and report Top-1 and mean accuracies.

**Evaluation Protocols.** We evaluate our model using three protocols:

- *Zero-Shot Transfer* assesses generalization on unseen data and tasks without extra tuning. We conduct zero-shot evaluation EgoMCQ, EK100-MIR, EK100-CLS, and EGTEA.
- *Visual & Textual Representation* is evaluated through EgoNLQ and EgoMC, where we use our pre-trained model as a visual/textual feature extractor. Following [4, 5], we train downstream models (VSLNet [15] for EgoNLQ, VSGN [14] for EgoMC) with pre-computed features.
- *Vision-Language Grounding.* We evaluate local entity understanding and interpretation via qualitative results on EgoCLIP [3]. We illustrate the saliency maps produced by our model and compare it with bounding boxes from [4].

Table 3: Ablation results on multi-grained losses.

| Loss Settings | | | | EgoMCQ | | EK100-MIR | | EK100-CLS | |
|---|---|---|---|---|---|---|---|---|---|
| $\mathcal{L}_{ego}$ | $\mathcal{L}_{NEC}$ | $\mathcal{L}_{VEC}$ | $\mathcal{L}_{proj}$ | Inter | Intra | Avg mAP | Avg nDCG | Top-1 Acc | Top-5 Acc |
| ✅ | ❌ | ❌ | ❌ | 93.4 | 58.4 | 30.8 | 32.7 | 18.2 | 36.8 |
| ✅ | ✅ | ❌ | ❌ | 93.6 | 59.9 | 30.9 | 32.8 | 19.1 | 37.5 |
| ✅ | ❌ | ✅ | ❌ | 93.7 | 59.7 | 31.1 | 32.9 | 18.9 | 37.3 |
| ✅ | ❌ | ❌ | ✅ | 93.2 | 58.5 | 30.8 | 32.6 | 18.5 | 37.0 |
| ✅ | ✅ | ✅ | ❌ | 94.0 | 61.1 | 31.3 | 33.0 | 19.3 | 37.7 |
| ✅ | ✅ | ✅ | ✅ | 94.1 | 61.3 | 31.3 | 33.1 | 19.3 | 38.2 |

## 5.3 Main Results

**Comparison in Zero-shot Transfer.** In Table 1, to ensure fairness, we re-train HelpingHands [4] using their official codebase with TSF-B backbone and the same pre-trained weights as ours, provided by LaViLa [5]. Our model consistently outperforms previous SOTA, achieving 3.1% improvement in top-1 accuracy on EK100-CLS, 0.5% and 0.3% increase in intra- and inter-video accuracy on EgoMCQ, and 0.5% improvement in mean accuracy on EGTEA. It also competes competitively with HelpingHands in the video/text retrieval EK100-MIR. Overall, our method demonstrates strong performance for zero-shot transfer across multiple benchmarks.

**Comparison in Visual & Textual Representation.** In Table 2, our method outperforms prior SOTA models across all metrics in EgoNLQ by adequate gaps. In EgoMQ, HENASY shows comparable performance, particularly excelling in mAP where it surpasses SOTA by 1%. This highlights HENASY's effectiveness when being applied to downstream models for features extraction.

**Vision-Language Grounding** We include qualitative experiment (Fig. 4) to compare with HelpingHands [4], which, in our knowledge, is the only VLM including weak visual grounding capacity via bounding boxes. As we can see, HENASY provides stronger interpretation with saliency maps reflecting dynamically evolving regions that most related to both appearance and motion queries. Furthermore, HelpingHands cannot correctly perform grounding with verb phrases (e.g., "scrolling the phone"), therefore, we show the bounding box of noun instead (e.g., "phone").

## 5.4 Ablation Studies

**Losses:** We investigate various combinations of multi-grained loss components and report results in Table 3. We found that HENASY trained only with instance-level loss $\mathcal{L}_{ego}$ yields 1-2% lower across all benchmarks compared to full loss setting. Besides, $\mathcal{L}_{VEC}$ contributes slightly more to performance gains in EgoMCQ and EK100-MIR, compared to $\mathcal{L}_{NEC}$. Finally, $\mathcal{L}_{proj}$ shows a slight improvement of the overall performance.

**Impact of Entity-Aware Decoder:** We evaluate the influence of the entity-aware (EA) decoder on zero-shot tasks in the first two rows of Table 4. In the first experiment (labeled as 'w/ avg. pool'), the proposed EA decoder is omitted, and entity tokens are merely processed using average pooling to generate the video representation. This configuration results in a notable decline in performance across all benchmarks. In the second experiment (labeled as 'w/

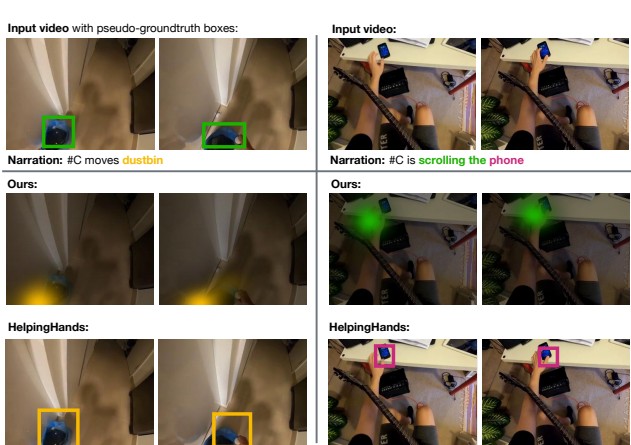

Figure 4: **Vision-Language Grounding.** Qualitative comparisons with HelpingHands [4] on EgoCLIP [3]. **Left:** comparison with a noun query obtained from narration and the pseudo-groundtruth boxes detected by [33] for reference. **Right:** verb phrase in the narration is used for comparison, as verb phrase cannot be captured by [33], we do not include pseudo boxes.

Table 4: Ablation results on model design.

| Model designs | EgoMCQ | | EK100-MIR | | EK100-CLS | |
|---|---|---|---|---|---|---|
| | Inter | Intra | Avg mAP | Avg nDCG | Top-1 Acc | Top-5 Acc |
| w/ avg. pool | 87.6 | 47.9 | 18.8 | 25.5 | 6.7 | 18.1 |
| w/ SA dec. | 93.3 | 59.1 | 30.4 | 32.8 | 18.0 | 36.3 |
| w/o bootstrap | 92.6 | 59.2 | 31.1 | 32.6 | 19.2 | 37.9 |
| **complete settings** | 94.1 | 61.3 | 31.3 | 33.2 | 19.5 | 38.2 |

Table 5: Comparison on computational complexity and memory cost.

| | HelpingHands | Ours |
|---|---|---|
| **Autoregressive** | ✔ | ✖ |
| GFLOPs per video clip (Million) | 530 | 599 |
| Number of Parameters (Million) | 216 | 291 |
| Train GPU Memory (GB) | 38 | 42 |
| Inference GPU Memory (GB) | 4.4 | 4.8 |
| Inference Time (seconds) | 2.87 | 1.02 |

SA Dec.'), video patch tokens are combined with entity tokens and then input into a self-attention decoder, which has the same dimensions as our EA decoder. This setup leads to a performance decrease of approximately 2% across benchmarks compared to our proposed decoder ('complete settings').

These ablation studies show that modeling interactions between global features and entity embeddings plays a critical role, and the proposed design of entity-aware decoder significantly enhances overall model performance.

**Impact of Bootstrapping Stage:** We report the effect of bootstrapping stage in the third row of Table 4 (labeled as w/o bootstrap), where we remove bootstrapping stage by directly processing video patch tokens. The performance degrades by 1% across all benchmarks, showing the effectiveness of this design choice.

**Computational and Memory Costs:** We compare our method with HelpingHands [4] in Table 5. Our model is slightly more expensive but quite competitive in terms of memory requirements, the number of parameters and GFLOPs. Importantly, our inference time is 3 times faster than that of the HelpingHands. This superior running time of HENASY compared to HelpingHands can be attributed to HelpingHands' utilization of an autoregressive decoder, which reduces parallel computations and makes it less efficient despite its lower computational cost.

## 6 Conclusions

In this work, we explored the Hierarchical Entities Assembly framework, dubbed HENASY, which is designed to improve video representation of previous vision-language models by addressing their limitations in fine-grained modeling. Our model explicitly captures the dynamic interactions between visual entities to form a comprehensive video representation. Our experiments showed that HENASY outperforms existing SOTA methods across challenging egocentric video understanding benchmarks like EgoMCQ, EK100-MIR, EK100-CLS, EgoNLQ, and EgoMQ in both zero-shot transfer and feature extraction settings, while also demonstrating strong interpretation capabilities. Despite these strengths, there are several opportunities for future work to improve our model further.

**Limitations and Future Works** Although our focus has been on tasks utilizing ViT encoders for a variety of benchmarks, we believe it is important to extend HENASY to generative tasks such as video generation (e.g., stable diffusion) or to handle long-form videos. While HENASY can provide interpretability by focusing on relevant scene entities for both objects and actions, it is still limited in explicitly showing the interactions between scene entities. This necessitates the development of a dynamic scene graph, which remains an open question due to the unavailability of data.

## Acknowledgements

The authors gratefully acknowledge funding supports from U.S. National Science Foundation (NSF) under Award No. OIA-1946391 and NSF EFRI BRAID 2223793.

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

## A  Divided Space-Time Block

Divided Space-Time (DST) block [32] is mainly utilized in the global encoder and entity grouping stage of the local entity encoder in our HENASY framework.

In global encoder, DST typically takes a concatenation of learnable CLS token and video patch tokens, i.e., $[\mathbf{c}^l; \mathbf{z}^l]$ as inputs. While in the local entity encoder, inputs to DST comprises segment tokens $\mathbf{g}^l_{entity}$ and segment tokens $\mathbf{s}^l$.

A DST block reduces computational cost of a full space-time attention by factorizing it into time and space attention, consecutively:

$$\tilde{\mathbf{y}}^l_{t,k} = \sum_{t'=1}^{T} \text{Softmax} \left\{ (\mathbf{q}^l_{t,k} \cdot \mathbf{k}^l_{t',k})/\sqrt{d_h} \right\} \mathbf{v}^l_{t',k}$$

$$\mathbf{y}^l_{t,k} = \sum_{k'=1}^{K} \text{Softmax} \left\{ (\tilde{\mathbf{q}}^l_{t,k} \cdot \tilde{\mathbf{k}}^l_{t,k'})/\sqrt{d_h} \right\} \tilde{\mathbf{v}}^l_{t,k'}$$

where $\mathbf{q}^l_{t,k}, \mathbf{k}^l_{t,k}, \mathbf{v}^l_{t,k} \in \mathbb{R}^{d_h}$ are query, key, and value vectors, respectively, which are linearly projected from the input of DST block after being split by number of heads. Likewise, $\tilde{\mathbf{q}}^l_{t,k}, \tilde{\mathbf{k}}^l_{t,k}, \tilde{\mathbf{v}}^l_{t,k} \in \mathbb{R}^{d_h}$ are query, key, and value vectors derived from $\tilde{\mathbf{y}}^l_{t,k}$.

## B  Temporal-Aware Grouping

### B.1  Details of Tokens Assignment and Grouping

**Similarity Computation.**

Given learnable group tokens $\mathbf{g}_q \in \mathbb{R}^{Q \times D}$ and input tokens to be grouped $\mathbf{i} \in \mathbb{R}^{T \times I \times D}$, we follow [12] to compute the 3D similarity array $\mathbf{A} \in \mathbb{R}^{T \times Q \times I}$ between each video-level group token $\mathbf{g}_i \in \mathbf{g}_q$ and every segment token $\mathbf{i}_{t,j} \in \mathbf{k}$, where $t$ and $j$ are temporal and spatial indices, respectively. Gumbel-Softmax [39] is then applied to rescale similarity matrices over group tokens:

$$\mathbf{A}_{t,i,j} = \frac{\exp\left(W_q \mathbf{g}^l_i \cdot W_i \mathbf{i}_{t,j} + \gamma_i\right)}{\sum_{k=1}^{Q} \exp\left(W_q \mathbf{q}_k \cdot W_i \mathbf{i}_{t,j} + \gamma_k\right)} \tag{11}$$

where $W_q$ and $W_i$ are learned linear projections for group and segment tokens, respectively, and $\gamma_i$ is sampled from $Gumbel(0,1)$ distribution.

**Group Assignment.** Afterwards, a segment token is hardly assigned to a group token via $\arg\max$ operation over group tokens (non-differentiable) with the straight-through trick [40] to allow end-to-end training:

$$\tilde{\mathbf{A}} = \text{one-hot}\left(\arg\max_i \mathbf{A}\right) + \mathbf{A} - sg(\mathbf{A}) \tag{12}$$

where $sg(\cdot)$ is a stop-gradient function, and one-hot$(\cdot)$ operator converts the assigned group indices into one-hot vectors.

### B.2  Saliency Map Generation

Saliency maps of each dynamic entity that evolving across frames of input video can be constructed from similarity arrays produced in temporal-aware grouping layers at bootstrapping and entity grouping stage. Let denote them as $\mathbf{A}^{boot}$ and $\mathbf{A}^{entity}$, respectively. We first compute the assignment probability array between video patches at each frame $t$ and final entity tokens by the following equation:

$$\mathbf{M}_t = \mathbf{A}^{boot}_t \cdot (\mathbf{A}^{entity}_t)^T \tag{13}$$

where $t$ is a frame index in $T$, and $\mathbf{M} \in \mathbb{R}^{T \times K \times E}$ ($K$ is the number of patches per frame). Then, saliency maps can be obtained via softmax activation function over the patches $\hat{\mathbf{M}} = \text{softmax}_K(\mathbf{M})$. Splitting the saliency array $\hat{\mathbf{M}}$ over entity dimension, we can obtain saliency maps of all frames, each of which highlights the spatial location and shapes of the corresponding entity.

## C    Verb Phrase Generation

We utilize Llama-2 [34] to generate verb phrases from narration due to its superior performance in processing free-form texts. Below is a prompt we design to capture verb phrases:

- System: "Act as if you are a robot that only outputs python list of strings."
- User: "Task: You are given an input sentence. Your job is to output the action verb phrases, which are always starting by a verb-ing."

## D    Quantitative evaluation on visual grounding

We conducted a rigorous quantitative analysis on Ego4D dataset to compare with the SOTA model of HelpingHands, as they only provide visual grounding results on this datasets. We create semantic segmentation labels by ourselves for 200 videos. The results are reported under mIoU metrics between visual grounding prediction with corresponding groundtruth, showing our model's superior visual grounding capability compared to HelpingHands:

| Model | mIoU |
|-------|------|
| HelpingHands [4] | 22.73% |
| Ours | 41.06% |

Table 6: Comparison with HelpingHands on visual grounding task.

**Discussion:** Despite being the SOTA in egocentric tasks and visual grounding task, HelpingHands [4] only provides coarse bounding boxes of objects, leading to much lower mIoU scores due to inadequate coverage of the target masks. In contrast, our model employs segmentation masks that closely align with the ground truth, resulting in higher mIoU scores, demonstrating superior visual grounding capabilities.

