# OpenReview forum: "HENASY: Learning to Assemble Scene-Entities for Interpretable Egocentric Video-Language Model"
_NeurIPS.cc/2024/Conference — NeurIPS 2024 poster_

### Official Review · Reviewer_4RbK · 2024-07-12

**Soundness:** 3
**Presentation:** 3
**Contribution:** 2
**Rating:** 6
**Confidence:** 3

**Summary:**

This paper proposed HENASY, a novel framework for learning egocentric video-language models where texts are grounded to visual scene entities. Its main idea is to use both global and local visual encoder to encode video features so that nouns and verb phrases in the paired text could be matched individually. The overall framework is optimized through contrastive losses while a projection loss is further introduced to improve the grounding quality. Experiments show that the proposed the method could achieve good performance on a wide range of egocentric video tasks including video/text retrieval, action recognition, multi-choice query, natural language query, and moments query.

**Strengths:**

1. This paper is overall well-written and easy to follow.

2. The problem addressed by this paper is important to the field. Learning grounded and interpretable egocentric video-language models has been seldomly studied in previous works, which might raise new research interests in this field.

3. The experiments are extensive where the proposed method has been evaluated on a wide range of egocentric video-text tasks through zero-shot transfer or learned video-text representations.

**Weaknesses:**

1. The performance improvement of the proposed method on most tasks are very marginal, compared to HelpingHands, which is the most related work to the paper. Given the fact that the design of  HelpingHands is much simpler than the proposed methods, I have concern on whether the complex components introduced in this paper is efficient.

2. Given the huge complexity of the proposed method, some important studies on the proposed design choices are missing. E.g., (1) the authors mentioned that using GroupViT to directly process input video patch tokens will diminishe performance but the results are not reported in the paper; (2) rather than using [5] to initialize the global encoder, have the authors tried other options, and what are the results? These help the reader to gain a better understanding of the robustness of the proposed method.

3. What about the scaling properties of the proposed method? Does it scale well with larger model sizes or more training data?

**Questions:**

Please refer to the Weakness section.

**Limitations:**

The authors have widely discussed the limitation of the proposed work and also its challenges in modeling the interactions between complex scene entities.

---

> ### Author Rebuttal · Authors · 2024-08-06
>
> We appreciate your acknowledgment of our well-written paper, the importance and novelty of our approach, and the extensive experiments.
>
> ## **1: Comparison with HelpingHands**
>
> We recap the key improvements of our work over HelpingHands as below
>
> | Property | Our Work | HelpingHands |
> | --- | --- | --- |
> | Method | Explicitly models video as dynamic entities (object-centric), capturing their interactions (action-centric) to form spatiotemporal interpretable, object/action-aware video representation | Implicitly induces object occurrence (nouns) on top of pre-extracted feature map via auxiliary task-specific (detection) head to form an object-aware video representation |
> | Fine-grained alignment | Spatiotemporal fine-grained alignments. Object appearance: Noun-entity alignment. Activity motion: verb-entities alignments | No alignment at granular level |
> | Interpretability | Strong, both spatial and temporal dimensions with object-centric and action-centric via dynamic entity assembly mechanism | None. |
> | Visual grounding | Strong, with spatiotemporal saliency maps of scene entities and relationships | Weak, only provide predicted bounding boxes |
> | Efficiency | faster than HelpingHands x3 times in inference | slower x3 times in inference due to autoregressive decoder |
>
> This discussion is included in Section 2, lines 87-93 of the submission.
>
> We highlight our improvements over HelpingHands across various metrics as below (Tables 1 and 2 in the paper):
>
> | Benchmark | Metric | HelpingHands | Our Method | Improvement |
> | --- | --- | --- | --- | --- |
> | EgoMCQ | Inter Acc | 93.2 | 94.1 | +0.9 |
> | EgoMCQ | Intra Acc | 58.8 | 61.3 | +2.5 |
> | EGTEA | Top-1 Acc | 35.3 | 35.9 | +0.6 |
> | EgoNLQ | mIoU @0.3 R5 | 20.4 | 21.5 | +1.1 |
> | EgoMQ | mAP | 11.7 | 12.4 | +0.7 |
>
> **Discussion:** Our primary objective extends beyond achieving high benchmark scores to advancing interpretability within video-language models. Our model significantly diverges from HelpingHands by emphasizing compositional video understanding. Additionally, it improves upon HelpingHands by focusing on interpretable representations. This shift towards interpretability represents a strategic choice aimed at impacting areas where understanding model decisions is critical.
>
> ## **2: Ablation studies**
>
> In Lines 159-161, we identified 3 major limitations in slot-attention (GroupViT) if we merely employ it:
> (i) direct process of video patch tokens from scratch, which neglects powerful video representations already learned by pre-trained models.
>
> (ii) originally proposed in image domain, making it unable to model dynamic entities with temporal information in videos.
>
> (iii) unable to model object-environment interactions, which is essential to capture spatiotemporal dynamics of the videos.
>
> We address these limitations through:
> (i) A novel bootstrapping stage: This aims to leverage the powerful pre-trained model for video patches encoding. The corresponding ablation study was in Table 3, row 3.
>
> (ii) Temporal-aware grouping (TAG): This targets preserving temporal dimensions, crucial for our hierarchical dynamic entity assembly approach. Given the foundational role of TAG in our model, we did not conduct a separate ablation study for it, as removing it would undermine the model’s functionality.
>
> (iii) Entity-aware decoder: This decoder plays a crucial role in ****modeling entities-environment interactions**,** enhancing our model’s overall performance. It propagates entity-level features from local entity encoder to enrich video-level embeddings from global encoder. It significantly contributes for the effective performance of our model. If we excluded this decoder, the model would rely solely on entity features from our slot-based local entity encoder and would end up with a substantial performance drop (Table 3, rows 1-2).
>
> ## **3: Initialization with other pre-trained models**
>
> Due to time constraint of the rebuttal period, we were able to conduct experiments on an additional pre-trained model, EgoVLP, besides the pre-trained model LaViLa reported in the submission. Below is a performance comparison when initializing our model with different pre-trained models, against the baseline results from LaViLa and EgoVLP:
>
> | Benchmark | Metric | LaViLa | Ours (initialized with LaViLa) | EgoVLP | Ours (initialized with EgoVLP) |
> | --- | --- | --- | --- | --- | --- |
> | EgoMCQ | Intra Acc | 59.9 | 61.3 | 57.2 | 59.7 |
> | EK100-MIR | Avg mAP | 30.9 | 31.3 | 23.3 | 30.9 |
> | EGTEA | Top-1 Acc | 35.5 | 35.9 | 17.6 | 34.0 |
>
> This table demonstrates that our model, when initialized with EgoVLP, not only performs competitively with its initialization using LaViLa, but also consistently outperforms the original EgoVLP across various benchmarks. This highlights the versatility and robustness of our model. We will add this result in our final version with an extra page.
>
> ## **4: Scalability**
>
> In our submission, we primarily used TSF-B as the backbone. As suggested by the reviewer, we have trained with a larger version, TSF-L, to further investigate the scalability. Due to time constraints, this model has not finished its training process. Nevertheless, we are pleased to report the current best results and comparison as below:
>
> |Benchmark|EK100-MIR|EK100-CLS| EGTEA | EgoMCQ |
> |---|---|---|---|---|
> | Metric | Avg mAP | Top-1 Acc | Top-1 Acc | Intra Acc |
> | LaViLa w/ TSF-B | 30.9 | 16.4 | 35.5 | 59.9 |
> | LaViLa w/ TSF-L | +5.2 | +4.5 | +4.6 | +3.2 |
> | Our w/ TSF-B | 31.3 | 19.5 | 35.9 | 61.3 |
> | Ours w/ TSF-L | +5.1 | +5.3 | +5.3 | +2.7 |
>
> This table illustrates the performance improvements of TSF-L over TSF-B for both LaViLa and our method. The results demonstrate that our model scales effectively with the larger backbone, showing consistent or greater gains across all benchmarks compared to LaViLa. These evidences underscore the scalability of our approach when equipped with more powerful backbones. We will add our final result to camera-ready version.

---

> > ### Author Response · Authors · 2024-08-12
> > **Follow Up Our Rebuttal**
> >
> > Dear Reviewer 4RbK
> >
> > We sincerely appreciate the time and effort you have dedicated to providing feedback on our work. Your insights are invaluable in helping us improve its clarity and overall quality. We want to follow up to check if our response fully addressed your concerns/questions before ending the discussion period.
> >
> > Thank you once again, and we look forward to your feedback.

---

> > > ### Comment · Reviewer_4RbK · 2024-08-12
> > >
> > > I thank the authors for their responses. Most of my concerns have been addressed. I just have one followup question on "2. Ablation studies" (on GourpViT): I acknowledge the proposed improvements on GroupViT, but I wonder did you have a chance to get the numbers if GroupViT is used. I am asking because my major concern about the work is still the additional complexity introduced by the method, therefore presenting the performance gain brought by the model itself should make the proposed method more convincing.

---

> > > > ### Author Response · Authors · 2024-08-12
> > > >
> > > > Dear reviewer 4RbK,
> > > >
> > > > Thank you for taking the time to review our response. We value your follow-up question regarding our use GroupViT. We appreciate your insights and the opportunity to further clarify our contributions and novelty over GroupViT.
> > > >
> > > > Initially, it is important to note that GroupViT was originally designed for static images and does not inherently support video processing. To address this limitation, we proposed a temporal-aware grouping (TAG) mechanism specifically adapted for video patches grouping. This mechanism is critical for our hierarchical dynamic entities grouping approach, hence, it cannot be removed.
> > > >
> > > > With such mechanism in place, we tried to leverage the original architecture of GroupViT as much as possible, including number of groups and stages (which are fixed in all reported experiments). Unfortunately, the experiment resulted in significantly poor performance. Let’s call this version as GroupViT+TAG, below are its performances on EK-MIR and EK-CLS datasets and comparisons to other ablation studies (Table 3):
> > > >
> > > > | Model | GroupViT | TAG | Bootstrapping stage | Entity-aware decoder | EK-MIR (Avg. mAP) | EK-MIR (Avg. nDCG) | EK-CLS (Top-1 Acc.) | EK-CLS (Top-5 Acc.) |
> > > > | --- | --- | --- | --- | --- | --- | --- | --- | --- |
> > > > | GroupViT+TAG | ✓ | ✓ |  |  | 17.2 | 25.2 | 6.2 | 17.8 |
> > > > | w/o entity-aware decoder | ✓ | ✓ | ✓ |  | 18.8 | 25.5 | 6.7 | 18.1 |
> > > > | w/o bootstrapping stage | ✓ | ✓ |  | ✓ | 31.1 | 32.6 | 19.2 | 37.9 |
> > > > | complete setting | ✓ | ✓ | ✓ | ✓ | 31.3 | 33.2 | 19.5 | 38.2 |
> > > >
> > > > We would like to note that the ablation study of GroupViT+TAG was conducted at the initial phase of our work.
> > > >
> > > > This empirical finding of the GroupViT+TAG led us to propose two critical enhancements: the bootstrapping stage and the entity-aware decoder. By integrating these modules, our model extends beyond the original capabilities of GroupViT.
> > > >
> > > > We hope this response would answer your concern and we welcome any further discussion you may have.

---

> > > > > ### Comment · Reviewer_4RbK · 2024-08-13
> > > > >
> > > > > Thanks for your detailed reply. I think the provided new results make the model design more convincing and encourage the authors to add them to the revised version of the paper. Given all of my concerns have been addressed, I would like to increase my rating to weak accept.

---

> > > > > > ### Author Response · Authors · 2024-08-13
> > > > > > **Appreciation for Your Revised Feedback and Acceptance**
> > > > > >
> > > > > > Thank you very much for your encouraging feedback and for taking the time to re-evaluate our work. We are delighted to hear that our responses and the new experiment have addressed your concerns effectively.
> > > > > >
> > > > > > We will certainly incorporate the new results into the final version of the paper when one additional page is allowed, as you recommended.
> > > > > >
> > > > > > We are grateful for your updated rating and truly appreciate your support for our work. Thank you once again for your thoughtful and thorough review.

---

### Official Review · Reviewer_cNiH · 2024-07-13

**Soundness:** 3
**Presentation:** 3
**Contribution:** 3
**Rating:** 5
**Confidence:** 3

**Summary:**

The paper introduces a novel framework for improving interpretability and performance in video-language models, specifically for egocentric videos. The framework, Hierarchical ENtities ASsemblY, employs a spatiotemporal token grouping mechanism that assembles and models relationships between dynamically evolving scene entities. This method aims to mimic human perceptual abilities by focusing on a compositional understanding of scene dynamics. The training of HENASY incorporates multi-grained contrastive losses, enhancing the model's ability to produce robust entity-level and video-level representations. Extensive experimental results demonstrate that HENASY significantly outperforms existing benchmarks on a range of egocentric video tasks.

**Strengths:**

1. HENASY introduces a groundbreaking approach to VLMs by focusing on dynamic scene entity assembly, which significantly diverges from traditional methods that generally emphasize static frame analysis.
2. The paper provides comprehensive experimental evidence showing that HENASY achieves superior performance on multiple egocentric video benchmarks, validating the effectiveness of its novel methodologies.
3. By leveraging visual grounding with free-form text queries, HENASY enhances the interpretability of VLMs, which is critical for applications requiring transparent decision-making processes.
4. The use of multi-grained contrastive losses to optimize the model at both the entity and video levels is a well-defined objective that contributes to the model's strong performance.

**Weaknesses:**

1. The paper lacks a detailed discussion on the scalability and computational demands of HENASY, which are crucial for its application in real-world settings, especially on resource-constrained devices.
2. While HENASY shows impressive results in controlled experiments, its ability to generalize across diverse real-world scenarios and different video domains remains underexplored.

**Questions:**

1. Could the authors provide insights into HENASY's performance on diverse real-world datasets and its computational efficiency and scalability compared to existing methods?
2. Can the authors elaborate on the spatiotemporal token grouping mechanism’s impact on training dynamics and model convergence, and provide a comparative analysis of interpretability with other state-of-the-art methods?

**Limitations:**

The paper should explicitly address potential limitations related to the scalability of HENASY and its performance on varied real-world datasets. Additionally, the authors should discuss any potential negative societal impacts, such as privacy concerns in the context of egocentric video analysis, and suggest possible mitigation strategies for such issues to ensure ethical deployment.

---

> ### Author Rebuttal · Authors · 2024-08-07
>
> We appreciate the reviewer of their acknowledgment that our approach is groundbreaking with strong interpretability, comprehensive experiments. We would like to address concerns and questions raised by the reviewer as follows:
>
> ## **1: Scalability and computation**
>
> **Scalability**: In our submission, we primarily reported results using TSF-B architecture as the backbone. As suggested by the reviewer, we have initiated training with a larger version of the backbone, TSF-L, to further investigate the scalability of our method. However, due to the constrained time frame of the rebuttal period, this model has not finished its training process. Nevertheless, we are pleased to report the current best results and comparison with a baseline (LaViLa) using similar backbones:
>
> | Benchmark | EK100-MIR | EK100-CLS | EGTEA | EgoMCQ |
> | --- | --- | --- | --- | --- |
> | Metric | Avg mAP | Top-1 Acc | Top-1 Acc | Intra Acc |
> | LaViLa w/ TSF-B | 30.9% | 16.4% | 35.5% | 59.9% |
> | LaViLa w/ TSF-L | +5.2% | +4.5% | +4.6% | +3.2% |
> | Our w/ TSF-B | 31.3% | 19.5% | 35.9% | 61.3% |
> | Ours w/ TSF-L | +5.1% | +5.3% | +5.3% | +2.7% |
>
> This table illustrates the performance improvements of TSF-L over TSF-B for both LaViLa and our method. The results demonstrate that our model scales effectively with the larger backbone, showing consistent or greater gains across all benchmarks compared to LaViLa. These evidences underscore the scalability of our approach when equipped with more powerful backbones. We will add our final result to camera-ready version.
>
> **Computation**: We have conducted a study to evaluate the computational cost (Table 5, Page 9 of the submission). In this study, we also compared with the SOTA HelpingHands. It is summarized as follows.
>
> |  | HelpingHands | Ours |
> | --- | --- | --- |
> | Autoregressive  | YES | NO |
> | GFLOPs (per clip) | 530M | 599M |
> | Number of Parameters | 216M | 291M |
> | GPU Memory (train) | 38GB | 42GB |
> | GPU Memory (inference) | 4.4GB | 4.8GB |
> | Inference Time (seconds) | 2.87 | 1.02 |
>
> ## **2: Resource-constrained devices**
>
> In real-world applications, computational resource constraints can vary significantly, leading us to categorize environments into two primary types: cloud computation and standalone systems. For cloud computation, HENASY is particularly well-suited as it can leverage extensive computational resources similar to those used in deploying large language models (LLMs). For standalone systems, we are investigating several optimization techniques to ensure efficient deployment. These techniques include model pruning, quantization, and knowledge distillation, which notably reduce model size and computational load while maintaining competitive performance. The original HENASY model requires 4.8GB of GPU memory, making it compatible with modern devices like the Jetson AGX Xavier (up to 32GB GPU memory), Jetson Xavier NX (8GB GPU memory), and Jetson TX2 (8GB GPU memory). However, deployment in resource-constrained environments is beyond the scope of this paper.
>
> ## **3: Generalize across diverse real-world scenarios and different video domain**
>
> As clarified in the session “Benchmarks and Evaluation Protocols” on Pages 7, 8 of the submission, we evaluated the proposed HENASY using different protocols, including  zero-shot transfer protocol and visual & textual representation protocol. We conducted evaluation on various datasets, including  EgoMCQ, EgoNLQ, EgoMQ, EK100-MIR, EK100-CLS, and EGTEA. It is important to note that  those datasets have been acquired by different devices under different real-world scenarios settings and across various video domains as follows:
>
> - EK-100: kitchen activities, collected by GoPro.
> - EGTEA: indoor activities, collected by SMI eye tracking glasses.
> - EgoMCQ, EgoNLQ, EgoMQ: indoor and outdoor daily activities (cooking, sport, shopping, lawn mowing, etc.), collected by multiple cameras such as GoPro, Vuzix Blade, Pupil Lab, etc.
>
> The experimental results in Table 1, Page 8 on the zero-shot transfer protocol and Table 2, Page 9 on the visual and textual representation protocol demonstrate the effectiveness of our proposed method across various video domains in diverse real-world scenarios. Notably, our model consistently outperforms previous SOTA methods with significant margins.
>
> ## **4: Impact of spatiotemporal token grouping mechanism**
>
> Our Temporal-aware grouping (TAG) mechanism has a foundational role in our model, which leverages slot-based methods in images into modeling spatiotemporal token grouping. Concretely, it helps preserve the temporal dimension during our hierarchical dynamic entities assembly process. Given the essential role of TAG in our model, we are unable to conduct a separate ablation study for it, as its removal would undermine the functionality of our model.
>
> ## **5: Comparative analysis of interpretability**
>
> We conduct a quantitative experiment on Ego4D for visual grounding task. In the absence of ground truth segmentation masks for Ego4D, we generated these masks ourselves. Due to time constraints, we managed to obtain labels for 200 videos. Below, we present the performance comparison in terms of mIoU scores with HelpingHands, our most related work that supports visual grounding:
>
> | Model | mIoU  |
> | --- | --- |
> | HelpingHands | 22.73% |
> | Ours | 41.06% |
>
> **Discussion:** although being the SOTA in egocentric tasks and visual grounding task, HelpingHands only provides coarse bounding boxes of objects, leading to lower mIoU scores due to inadequate coverage of the target masks. In contrast, our model employs segmentation masks that closely align with the ground truth, resulting in higher mIoU scores,  demonstrating superior visual grounding capabilities.

---

> > ### Author Response · Authors · 2024-08-12
> > **Follow Up Our Rebuttal**
> >
> > Dear Reviewer cNiH
> >
> > We sincerely appreciate the time and effort you have dedicated to providing feedback on our work. Your insights are invaluable in helping us improve its clarity and overall quality. We want to follow up to check if our response fully addressed your concerns/questions before ending the discussion period.
> >
> > Thank you once again, and we look forward to your feedback.

---

> > > ### Author Response · Authors · 2024-08-13
> > > **Kindly Follow-up on Rebuttal**
> > >
> > > Dear Reviewer cNiH,
> > >
> > > As we approach the end of the discussion period, we wanted to kindly remind you of our rebuttal. We are keen to ensure that our responses have thoroughly addressed your concerns and would appreciate any additional feedback you may have.
> > >
> > > Thank you once again for your invaluable insights and the time you have invested in reviewing our work. We look forward to hearing from you soon.

---

> ### Author Response · Authors · 2024-08-14
> **Reminder: Feedback Request Before Discussion Period Closes**
>
> Dear Reviewer cNiH,
>
> As we are nearing the end of the discussion phase, we kindly seek your feedback to ensure that our responses have addressed your initial concerns. If our rebuttal satisfactorily addressed your concerns and questions, we would be very grateful if you would consider to raise your rating.
>
> Thank you very much for your attention.

---

### Official Review · Reviewer_xaMC · 2024-07-21

**Soundness:** 3
**Presentation:** 2
**Contribution:** 2
**Rating:** 4
**Confidence:** 4

**Summary:**

The paper introduced HENASY, a novel framework for enhancing video-language understanding in egocentric videos. It utilized multi-grained contrastive losses from alignments of  video-narration, noun-entity, and verb-entity, to improve interpretability and performance. The method showed competitive results across various downstream tasks such as video-text retrieval, action recognition and visual question answering.

**Strengths:**

* The paper was well organized and easy to follow.
* This paper leveraged more fine-grained information on egocentric video-language models apart from the original instance-level alignment. In detail, it included noun-entity and verb-entity alignment as a bonus.
* The framework achieved competitive results in various downstream tasks, such as video/text retrieval and action recognition, showcasing its effectiveness in real-world applications compared with other pre-trained models.

**Weaknesses:**

* Such an idea of leveraging fine-grained information, especially alignment of nouns/verbs and the video contents, has been intensively studies in non-egocentric scenarios and there are a number of related papers [1, 2]. It is not clear whether there is a significant different in egocentric videos. Otherwise, it is expected to improve the model performance by incorporating those fine-grained alignments in the pre-training stage.
* Even though the structure to extract noun/verb entities was a little different, I feel like the essential idea was still similar to GroupViT. The authors mentioned that if directly using GroupViT, the performance would drop. Was the comparison in Table 3, w/o bootstrapping? Actually, in GroupViT there was also a hierarchical design to use different query tokens at different stages. Then what's the main contribution of the design in the paper? I am wondering if simply adding the number of trainable parameters of GroupViT to the same number in the paper would lead to comparable performance.
* I am wondering if co-training with data mixes from both egocentric and non-egocentric videos would boost performance for both domains. In addition, can the model pre-trained on the egocentric scenario be transferred to the non-egocentric tasks?
* The evaluation on visual grounding was not sufficient. Currently there were only some qualitative results in Figure 4. It would be better to report some quantitative results for more concrete tasks.

[1] Ge, Yuying, et al. "Bridging video-text retrieval with multiple choice questions." CVPR 2022.

[2] Xiong, Yuanhao, et al. "Structured Video-Language Modeling with Temporal Grouping and Spatial Grounding." ICLR 2024

**Questions:**

My questions are listed in the part of weaknesses above.

**Limitations:**

They had a separate "Limitations" section and elaborated on future works that could improve the model.

---

> ### Author Rebuttal · Authors · 2024-08-07
>
> We thank the reviewer for recognizing that our paper is well-organized, easy to follow, competitive results in real-world applications.
>
> ## **1: Comparison with [1, 2]**
>
> We provide the comparison as below:
>
> | Property | Ours | [1] | [2] |
> | --- | --- | --- | --- |
> | Method | Explicitly models video as dynamic entities (object aware), capturing their interactions (action aware)  | Employs a proxy module (BridgeFormer) to implicitly inject fine-grained information of text into video representation during training | Explicitly models video via group tokens (object aware). It is unable to capture the  interactions between objects (action aware) |
> | Fine-grained alignment | Spatiotemporal fine-grained alignments. Object appearance: Noun-entity alignment. Activity motion: verb-entities alignments | No granular level alignment | Only alignment on object appearance: Noun-entity alignment |
> | Interpretability | Strong, both spatial and temporal dimensions with object-centric and action-centric via dynamic entity assembly mechanism | None, the proxy module is removed during inference, rendering the model incapable of interpretation | Medium, only object-centric |
> | Visual grounding | Strong, with spatiotemporal saliency maps of scene entities and relationships | None | Weaker, only saliency map related to a scene entities |
> | Efficiency | 2x A6000 GPU for training | 40x A6000 GPUs for training | No source code nor report |
>
> We note that [2] is published at ICLR 2024 (2 weeks before NeurIPS submission).
>
> ## **2: Comparison with GroupViT**
>
> To illustrate the challenges with merely applying GroupViT to video, how our method addresses these issues, and what set us stood out from GroupViT, we provide the following table:
>
> | Limitation in GroupViT | Our Innovative Solution |
> | --- | --- |
> | Domain: image | Domain: video |
> | Direct Processing of Patch Tokens: GroupViT learns to process input patches from scratch, which prevents it from utilizing the powerful representation of existing pre-trained models | Bootstrapping Stage: incorporates early layers of global encoder (using a frozen model from LaViLa) to leverage its powerful video representation. The effectiveness of this design choice is reported in our ablation (Table 3, row 3) |
> | Inability to Model Dynamic Entities: GroupViT is primarily proposed to process static images and is prohibitively unable to group video patches into dynamic entities | Temporal-Aware Grouping: a novel mechanism that merges  semantically similar tokens into larger entities while preserving temporal dimension of the video |
> | Absence of Object-Environment Interactions: GroupViT solely focuses on object level and lacks a mechanism to model relationships between objects and environment, which is crucial for spatiotemporal tasks in video domain. | Entity-Aware Decoder: a pivotal component that propagates entity-level features from local entity encoder to enrich video-level embedding in global encoder. It captures the object-environment interactions, enhancing the spatiotemporal representation. The effectiveness of entity-aware decoder is shown in our ablation (Table 3, rows 1-2) |
>
> ## **3: Co-training from both egocentric (ego) and exocentric (exo) and transferability from ego to exo tasks**
>
> On one hand, it is obvious that incorporating ego and exo is essential for many real applications in robotics and augmented reality. However, the dramatically different viewpoints pose significant challenges. Most ego datasets just recently become available, and very few ego videos are synchronized with corresponding exo videos. Additionally, ego datasets are much smaller in scale compared to exo ones. Therefore, leveraging exo to improve model performance on ego is a viable approach [A]. Early work [B] focused on representation learning using paired ego-exo videos. However, paired videos are much harder to obtain compared to unpaired ones. Recent studies [C] made notable progress with unpaired videos.
>
> On the other hand, transferability from ego to exo tasks presents a significant challenge and opportunity. The primary challenge is the alignment of different viewpoints, particularly between unpaired multi-view [D]. The ongoing research on joint view-invariant learning [E], domain adaptation [F], knowledge distillation [G] offer promising solutions.
>
> Despite co-training with mixed ego-exo data and transferability between ego-exo are exciting directions, our current objectives are solely focused on ego tasks, with an emphasis on providing trustworthiness and transparency to the model at a fine-grained level of alignment. Furthermore, we strongly believe that this capability can help bridge the gap between different viewpoints at fine-grained alignment, and this should be a focus of future research.
>
> [A] Weinland, D., etc "Making action recognition ...", ECCV2010
>
> [B] Yu, H., etc “What i see is what you see …”, ICM2019
>
> [C] Huang, Y., etc “EgoExoLearn …”, CVPR2024
>
> [D] Wang, Q., etc. “Learning from semantic alignment…”, ICCV2023
>
> [E] Xue, Z.S., etc. “Learning fine-grained …”, NeurIPS2023
>
> [F] Choi, J.,  etc, “Unsupervised and semi-supervised …”, WACV2020
>
> [G] Li, Y., etc. “Ego-exo…”, CVPR2021
>
> ## **4: Visual grounding**
>
> We conduct a quantitative experiment on Ego4D, as HelpingHands provides visual grounding results exclusively on this dataset. In the absence of ground truth segmentation masks for Ego4D, we generated these masks ourselves. Due to time constraints, we managed to obtain labels for only 200 videos. Below, we present the performance comparison in terms of mIoU scores:
>
> | Model | mIoU |
> | --- | --- |
> | HelpingHands | 22.73% |
> | Ours | 41.06% |
>
> **Discussion:** HelpingHands uses coarse bounding boxes for visual grounding, leading to lower mIoU scores due to inadequate coverage of the target masks. In contrast, our model employs segmentation masks that closely align with the ground truth, resulting in higher mIoU scores,  demonstrating superior visual grounding capabilities.

---

> ### Author Response · Authors · 2024-08-12
> **Follow Up Our Rebuttal**
>
> Dear Reviewer xaMC,
>
> We sincerely appreciate the time and effort you have dedicated to providing feedback on our work. Your insights are invaluable in helping us improve its clarity and overall quality. We want to follow up to check if our response fully addressed your concerns/questions before ending the discussion period.
>
> Thank you once again, and we look forward to your feedback.

---

> > ### Author Response · Authors · 2024-08-13
> > **Kindly Follow-up Our Rebuttal**
> >
> > Dear Reviewer xaMC,
> >
> > As we approach the end of the discussion period, we wanted to kindly remind you of our rebuttal. We are keen to ensure that our responses have thoroughly addressed your concerns and would appreciate any additional feedback you may have.
> >
> > Thank you once again for your invaluable insights and the time you have invested in reviewing our work. We look forward to hearing from you soon.

---

> ### Author Response · Authors · 2024-08-14
> **Reminder: Feedback Request Before Discussion Period Closes**
>
> Dear Reviewer xaMC,
>
> As we are nearing the end of the discussion phase, we kindly seek your feedback to ensure that our responses have addressed your concerns. If our rebuttal satisfactorily addressed your concerns and questions, we would be very grateful if you would consider to raise your rating.
>
> Thank you very much for your attention.

---

### Official Review · Reviewer_EqYJ · 2024-07-22

**Soundness:** 3
**Presentation:** 2
**Contribution:** 2
**Rating:** 5
**Confidence:** 5

**Summary:**

This paper presents HENASY (Hierarchical ENtities ASsemblY), a pretraining framework to learn scene-entities representations for egocentric videos. The authors proposed to learn compositional and hierarchal video representations by three levels: 1) global video features from a global video encoder. 2) entity features by assembling dynamic entities from video patches via local entity encoder, 3) interactions between entities and global context by an entity-aware decoder. With the designed model structure and losses, the authors conducted experiments on several egocentric tasks on two datasets. The results show the effectiveness of the proposed method.

**Strengths:**

1. The idea is well-motivated. The analysis in the introduction part that current simple instance-level video-caption alignment being hard to capture the complex and dynamic interactions among arbitrary entities are inspiring. The proposed method of utilizing both global information and local entitiy-level info is novel.
2. The authors conducted extensive experimets across multiple tasks including zero-shot transfer, down-stream finetuning and vision-language grounding. The experiments result show the effectiveness of the proposed method.
3. The ablation study (table 3 and 4) show the contribution of each proposed components clearly, providing intuiation and experience for future model design.

**Weaknesses:**

1. My major concern is the performance shown in the downstream task. In table 1, though the proposed methods generally achieved better performance compared with previous sota models (HelpingHands), the improvement seems to be minor. Besides, the paper didn't report the error analysis to dimish the influcen of randomness, which makes the contribution and effectiveness of the proposed method less convincing.
2. In Projection loss calculation, how is the mask predicted is not clearly presented.
3. The writing could be further improved. In line 127 it should be $L_{ego} = L_{ego}^{v2t} + L_{ego}^{t2v}$. In section 4, too many variables are used including z, c, g, s, e, making it hard to follow the context while reading.

**Questions:**

See weakness section.

**Limitations:**

Yes

---

> ### Author Rebuttal · Authors · 2024-08-06
>
> We thank the reviewer for their thoughtful evaluation and acknowledgment that our approach is well-motivated, novel, and effective under extensive experiments, providing intuition and experience for future model design. We will fix all noticed typos in our final version. Below, we would like to address the concerns raised:
>
> ## **1: Clarify the improvement over most recent SOTA model (HelpingHands)**
>
> We would like to recap the differences between our work and HelpingHands by the following table:
>
> | Property | Our Work | HelpingHands |
> | --- | --- | --- |
> | Method | Explicitly models video as dynamic entities (object-centric), capturing their interactions (action-centric) to form spatiotemporal interpretable, object/action-aware video representation | Implicitly induces object occurrence (nouns) on top of pre-extracted feature map via auxiliary task-specific (detection) head to form an object-aware video representation |
> | Fine-grained alignment | Spatiotemporal fine-grained alignments. Object appearance: Noun-entity alignment. Activity motion: verb-entities alignments | No alignment at granular level |
> | Interpretability | Strong, both spatial and temporal dimensions with object-centric and action-centric via dynamic entity assembly mechanism | None. |
> | Visual grounding | Strong, with spatiotemporal saliency maps of scene entities and relationships | Weak, only provide predicted bounding boxes |
> | Efficiency | faster than HelpingHands x3 times in inference | slower x3 times in inference due to autoregressive decoder |
>
> This discussion is included in Section 2, lines 87-93 of the submission.
>
> In addition, we highlight our improvements over HelpingHands across various metrics as below (Tables 1 and 2 in the paper):
>
> | Benchmark | Metric | HelpingHands | Our Method | Improvement |
> | --- | --- | --- | --- | --- |
> | EgoMCQ | Intra Acc | 93.2 | 94.1 | +0.9 |
> | EgoMCQ | Inter Acc | 58.8 | 61.3 | +2.5 |
> | EGTEA | Top-1 Acc | 35.3 | 35.9 | +0.6 |
> | EgoNLQ | mIoU @0.3 R5 | 20.4 | 21.5 | +1.1 |
> | EgoMQ | mAP | 11.7 | 12.4 | +0.7 |
>
> **Discussions:** We emphasize that the primary goal of our proposed method is to advance interpretable video representation by leveraging a compositional video understanding approach. This would create a new vibe to our research community from solely competing on benchmarks to prioritizing trustworthiness, transparency, and reliability. Although the improvements in some metrics may appear incremental, they are substantial within the context of interpretability and compositional analysis, as it pursuits of building transparent and accountable AI systems. Equipping the model with interpretability to ensure trustworthiness while enhancing performance is crucial for practical applications, especially in sensitive areas such as healthcare and autonomous driving.
>
> ## **2: Error Analysis to diminish the influence of randomness**
>
> Reproducibility is highly prioritized in our work, and as such, we have implemented fixed seeds in all random-related functions for every experiment to ensure that the community can reliably reproduce the performance reported in our paper.
>
> In response to this concern, we initiated evaluations of our model trained with different random seeds. However, due to the limited time available for the rebuttal and resource constraints, we can only train two additional models. We report the performance in terms of mean and std across several zero-shot benchmarks as below:
>
> | Benchmark | Metric | Our performance (mean±std) |
> | --- | --- | --- |
> | EK100-CLS | Top-1 Acc | 19.20 ± 0.51 |
> | EK100-MIR | Avg mAP | 31.07 ± 0.25 |
> | EGTEA | Top-1 Acc | 35.5 ± 1.1 |
> | EgoMCQ | Inter Acc | 59.74 ± 0.21 |
>
> The above table with high mean and low std indicates that our method is effectiveness, stable, robust and converges well under randomness.
>
> ## **3: Formulation of masks prediction for Projection loss**
>
> In Appendix Section B, we provided a comprehensive description of three steps involving the formulation of the masks prediction. Specifically:
>
> 1. **Similarity Computation**: We calculate a similarity array between the learnable group tokens and the input tokens.
> 2. **Group Assignment**: Each input token is assigned to a group token based on the highest similarity score. To retain differentiability essentially for training, we employ the straight-through trick during the assignment step. Obtaining assignment array.
> 3. **Saliency Map Generation**: Saliency maps are then generated using the assignment. These maps effectively highlight the spatial locations and dynamic contours of entities across different frames of the video, providing a visual representation of the model’s focus and understanding of the scene.
>
> The complete mathematical formulation and additional details are thoroughly explained in Appendix Section B, ensuring transparency and reproducibility of our methods. We will add reference to corresponding appendix in the projection loss description to make it clearer.
>
> ## **4: Clarification on the use of multiple variables**
>
> We use abbreviation to name variables for clarity and succinctness in mathematical formulations.
>
> To recap, there are five types of tokens as follows:
>
> - z: video patch tokens, which capture local features from video frames.
> - c: learnable <CLS> token, which is used to aggregate and represent global video features.
> - g: learnable group tokens, which are pivotal in organizing video information into meaningful clusters.
> - s: segment tokens, which are derived from segmenting video patch tokens after the bootstrapping stage.
> - e: entity tokens, which represent distinct entities clustered at the final stage of the process.
>
> To aid readability, we plan to include a notation table in the appendix, summarizing all variables and functions with clear descriptions and references to their respective sections. This will facilitate easier navigation and understanding of the methodology for all readers.

---

> > ### Comment · Reviewer_EqYJ · 2024-08-11
> >
> > Thank you for the authors' responses. I think the responses generally resolve my concerns and I decide to raise my score to weak accept.

---

> ### Author Response · Authors · 2024-08-12
> **Appreciation**
>
> Dear Reviewer EqYJ,
>
> Thank you for taking the time to read our rebuttal and for providing a positive rating. We greatly appreciate the valuable feedback you have given us, and we will incorporate your suggestions in the final revision of our paper.
>
> Thank you once again.

---

> > ### Author Response · Authors · 2024-08-13
> > **Kindly Follow-up on Review Rating Update**
> >
> > Dear Reviewer EqYJ,
> >
> > Thank you once again for your constructive feedback and for the time you have invested in reviewing our submission. We greatly appreciate your decision to raise your rating to “*weak accept*”.
> >
> > We noticed that the updated rating appears as “*borderline accept*” in the system. Therefore, we would like to kindly follow up with you regarding the rating update. We apologize for any inconvenience this request may cause and truly value your support and understanding. Thank you for your attention to this matter.

---

> > > ### Author Response · Authors · 2024-08-14
> > > **Kindly Reminder before Discussion Phase Ends**
> > >
> > > Dear Reviewer EqYJ,
> > >
> > > We apologize for the urgency of this message and the multiple reminders. As we approach the final hours of the discussion phase, we kindly request your assistance in updating the rating to reflect your previously indicated preference for a "weak accept". We appreciate your efforts and understanding in resolving this matter swiftly.
> > >
> > > We would like to thank you again for your invaluable feedback and consideration to raise your rating.

---

### Author Rebuttal · Authors · 2024-08-07

We sincerely appreciate all reviewers for their valuable time and constructive feedback. We are grateful for their recognition about the **significance, inspiration and impact** of our grounded and **interpretable** work (Reviewers EqYJ, cNiH, 4RbK), the **novelty** of our method (Reviewers EqYJ, cNiH), **extensive results** (Reviewers EqYJ, xaMC, cNiH), and **easy to follow presentation** (Reviewers xaMC, 4RbK).

The reviewers raised several concerns, which we addressed in the individual response. We summarize the highlights of our responses as follows:

**1. Architectural comparisons to related works [1, 2] and HelpingHands [3]:** We discuss the distinctions and advantages of our approach over [1, 2, 3], emphasizing the superior interpretability and dynamic entity modeling of our model. These properties are achieved via our compositional approach, which explicitly models video as dynamic entities (object-centric) and capturing their interactions (action-centric) to form spatiotemporal interpretable, object/action-aware video representation. Moreover, our approach is **the first to** explicitly integrate **spatiotemporal fine-grained alignments**, such as *object appearance (n*oun-entity alignment) and *activity motion* (verb-entities alignments).

**2. Performance Comparisons to HelpingHands [3]:** In Tables 1 and 2 of our submission, we reported significant improvements over HelpingHands, the most recent SOTA method, across various benchmarks and metrics. We would like to highlight again the such improvements that our model achieves against HelpingHands:

| Benchmark | Metric | HelpingHands | Our Method | Improvement |
| --- | --- | --- | --- | --- |
| EgoMCQ | Intra Acc | 93.2 | 94.1 | +0.9 |
| EgoMCQ | Inter Acc | 58.8 | 61.3 | +2.5 |
| EGTEA | Top-1 Acc | 35.3 | 35.9 | +0.6 |
| EgoNLQ | mIoU @0.3 R5 | 20.4 | 21.5 | +1.1 |
| EgoMQ | mAP | 11.7 | 12.4 | +0.7 |

**3. Comparison with GroupViT:** We clearly demonstrate three critical limitations inherent to GroupViT’s architecture:

(i) direct process of video patch tokens from scratch, which neglects powerful video representations already learned by pre-trained models.

(ii) originally proposed in image domain, making it unable to model dynamic entities with temporal information in videos.

(iii) unable to model object-environment interactions, which is essential to capture spatiotemporal dynamics of the videos.

In our local entity encoder, we address these limitations through:

(i) A novel bootstrapping stage: This aims to **leverage the powerful pre-trained model** for video patches encoding at early layers. The corresponding ablation study was in Table 3, row 3.

(ii) Temporal-aware grouping (TAG): This targets **preserving temporal dimensions**, crucial for our hierarchical dynamic entity assembly approach. Given the foundational role of TAG in our model, we did not conduct a separate ablation study for it, as removing it would undermine the model’s functionality.

(iii) Entity-aware decoder: This decoder plays a crucial role in **modeling entities-environment interactions,** enhancing our model’s overall performance. It propagates entity-level features from local entity encoder to enrich video-level embeddings from global encoder. It significantly contributes for the effective performance of our model. If we excluded this decoder, the model would rely solely on entity features from our slot-based local entity encoder and would end up with a substantial performance drop (Table 3, rows 1-2).

**4. Quantitative evaluation on visual grounding:** We conduct a rigorous quantitative analysis on Ego4D dataset to compare with the SOTA model of HelpingHands, as they only provide visual grounding results on this datasets. We create semantic segmentation labels by ourselves for 200 videos. The results are reported under mIoU metric between visual grounding prediction with corresponding groundtruth, showing our model’s superior visual grounding capability compared to HelpingHands:

| Model | mIoU |
| --- | --- |
| HelpingHands | 22.73% |
| Ours | 41.06% |

Our model outperforms HelpingHands thanks to its strong interpretability with dynamic segmentation masks associated with every entity in the video, while HelpingHands is only able to obtain bounding boxes of objects.

**5. Scalability:** We show that scaling of the network from the base model (TSF-B) to a large model (TSF-L) leads to the consistent gains in all benchmarks.

| Benchmark | EK100-MIR | EK100-CLS | EGTEA | EgoMCQ |
| --- | --- | --- | --- | --- |
| Metric | Avg mAP | Top-1 Acc | Top-1 Acc | Intra Acc |
| LaViLa w/ TSF-B | 30.9% | 16.4% | 35.5% | 59.9% |
| LaViLa w/ TSF-L | +5.2% | +4.5% | +4.6% | +3.2% |
| Our w/ TSF-B | 31.3% | 19.5% | 35.9% | 61.3% |
| Ours w/ TSF-L | +5.1% | +5.3% | +5.3% | +2.7% |

Finally, we have carefully addressed all the reviewers' comments and questions. We will revise and update the final version based on all suggestions using the allowed extra page.

[1] Ge, Yuying, et al. "Bridging video-text retrieval with multiple choice questions." CVPR 2022.

[2] Xiong, Yuanhao, et al. "Structured Video-Language Modeling with Temporal Grouping and Spatial Grounding." ICLR 2024.

[3] Zhang, Chuhan, et al. "Helping Hands: An Object-Aware Ego-Centric Video Recognition Model.” ICCV 2023.

---

### Author Response · Authors · 2024-08-11
**Thank you for your valuable comments !**

We would like to extend our gratitute once again to the reviewers for their constructive feedback and insightful questions, which have significantly helped us enhance the quality of our paper. We have diligently addressed all questions and suggestions from the reviewers in our rebuttal, including conducting new experiments as recommended.

We welcome any additional comments or insights the reviewers may have and are committed to promptly addressing further questions that arise.

In light of the rebuttal and additional data presented, we respectfully request that the reviewers reconsider the evaluation score for our paper. Thank you very much for your thoughtful consideration !

---

### Decision · Program_Chairs · 2024-09-25

**Decision:**

Accept (poster)

**Comment:**

This paper introduces a novel framework for egocentric video language understanding. The authors proposed to learn compositional and hierarchal video representations by three levels: 1) global video features from a global video encoder. 2) entity features by assembling dynamic entities from video patches via local entity encoder, 3) interactions between entities and global context by an entity-aware decoder. The paper received mixed reviews with 2 borderline accept, 1 weak accept and 1 borderline reject. Most reviewers see the merits of the paper: 1) clear motivation 2) extensive experiments on multiple downstream tasks 3) well-written. There are a few concerns raised by reviewers as well. Reviewer EqYJ and 4RbK think the improvement over Helping hands is marginal; Reviewer xaMC and 4RbK want to understand the delta between this work and group ViT; Reviewer cNiH and Reviewer cNiH question about scalability. Authors have provided detailed answers to reviewers' questions in their rebuttal. In specific, authors provided clear architectural comparison, more experiment results based on different base models, and comparison with GroupViT. Two reviewers acknowledged that their questions are resolved and increased their final scores. Considering all reviews and discussion, the AC recommends to accept this paper. The AC also encourages the authors to include the additional experiment results in their final version.